# CROSS-MODEL BACK-TRANSLATED DISTILLATION FOR UNSUPERVISED MACHINE TRANSLATION

## ABSTRACT

Recent unsupervised machine translation (UMT) systems usually employ three main principles: initialization, language modeling and iterative back-translation, though they may apply them differently. Crucially, iterative back-translation and denoising auto-encoding for language modeling provide data diversity to train the UMT systems. However, the gains from these diversification processes has seemed to plateau. We introduce a novel component to the standard UMT framework called Cross-model Back-translated Distillation (CBD), that is aimed to induce another level of data diversification that existing principles lack. CBD is applicable to all previous UMT approaches. In our experiments, it boosts the performance of the standard UMT methods by 1.5-2.0 BLEU. In particular, in WMT'14 English-French, WMT'16 German-English and English-Romanian, CBD outperforms cross-lingual masked language model (XLM) by 2.3, 2.2 and 1.6 BLEU, respectively. It also yields 1.5–3.3 BLEU improvements in IWSLT English-French and English-German tasks. Through extensive experimental analyses, we show that CBD is effective because it embraces data diversity while other similar variants do not.[1]

## 1 INTRODUCTION

Machine translation (MT) is a core task in natural language processing that involves both language understanding and generation. Recent neural approaches (Vaswani et al., 2017; Wu et al., 2019) have advanced the state of the art with near human-level performance (Hassan et al., 2018). However, they continue to rely heavily on large parallel data. As a result, the search for unsupervised alternatives using only monolingual data has been active. While Ravi & Knight (2011) and Klementiev et al. (2012) proposed various unsupervised techniques for statistical MT (SMT), Lample et al. (2018a;c) established a general framework for modern unsupervised MT (UMT) that works for both SMT and neural MT (NMT) models. The framework has three main principles: model initialization, language modeling and iterative back-translation. Model initialization bootstraps the model with a knowledge prior like word-level transfer (Lample et al., 2018b). Language modeling, which takes the form of denoising auto-encoding (DAE) in NMT (Lample et al., 2018c), trains the model to generate plausible sentences in a language. Meanwhile, iterative back-translation (IBT) facilitates cross-lingual translation training by generating noisy source sentences for original target sentences. The recent approaches differ on how they apply each of these three principles. For instance, Lample et al. (2018a) use an unsupervised word-translation model (Lample et al., 2018b) for model initialization, while Conneau & Lample (2019) use a pretrained cross-lingual masked language model (XLM).

In this paper, we focus on a different aspect of the UMT framework, namely, its *data diversification*[2] process. If we look from this view, the DAE and IBT steps of the UMT framework also perform some form of data diversification to train the model. Specifically, the noise model in the DAE process generates *new*, but noised, versions of the input data, which are used to train the model with a reconstruction objective. Likewise, the IBT step involves the same UMT model to create synthetic parallel pairs (with the source being synthetic), which are then used to train the model. Since the NMT model is updated with DAE and IBT simultaneously, the model generates *fresh* translations in each back-translation step. Overall, thanks to DAE and IBT, the model gets better at translating

---

[1] Anonymized code: https://tinyurl.com/y2ru8res.
[2] By diversification, we mean sentence level variations (not expanding to other topics or genres).

by iteratively training on the newly created and diversified data whose quality also improves over time. This argument also applies to statistical UMT, except for the lack of the DAE (Lample et al., 2018c). However, we conjecture that these diversification methods may have reached their limit as the performance does not improve further the longer we train the UMT models.

In this work, we introduce a fourth principle to the standard UMT framework: Cross-model Back-translated Distillation or CBD (§3), with the aim to induce another level of diversification that the existing UMT principles lack. CBD initially trains two UMT agents (models) using existing approaches. Then, one of the two agents translates the monolingual data from one language $s$ to another $t$ in the first level. In the second level, the generated data are back-translated from $t$ to $s$ by the *other agent*. In the final step, the synthetic parallel data created by the first and second levels are used to distill a supervised MT model. CBD is applicable to any existing UMT method and is more efficient than ensembling approaches (Freitag et al., 2017) (§5.3).

In the experiments (§4), we have evaluated CBD on the WMT'14 English-French, WMT'16 English-German and WMT'16 English-Romanian unsupervised translation tasks. CBD shows consistent improvements of 1.0-2.0 BLEU compared to the baselines in these tasks. It also boosts the performance on IWSLT'14 English-German and IWSLT'13 English-French tasks significantly. In our analysis, we explain with experiments why other similar variants (§5.1) and other alternatives from the literature (§5.4) do not work well and cross-model back-translation is crucial for our method. We further demonstrate that CBD enhances the baselines by achieving greater diversity as measured by back-translation BLEU (§5.2).

## 2 BACKGROUND

Ravi & Knight (2011) were among the first to propose a UMT system by framing the problem as a *decipherment* task that considers non-English text as a cipher for English. Nonetheless, the method is limited and may not be applicable to the current well-established NMT systems (Luong et al., 2015; Vaswani et al., 2017; Wu et al., 2019). Lample et al. (2018a) set the foundation for modern UMT. They propose to maintain two encoder-decoder networks simultaneously for both source and target languages, and train them via denoising auto-encoding, iterative back-translation and adversarial training. In their follow-up work, Lample et al. (2018c) formulate a common UMT framework for both PBSMT and NMT with three basic principles that can be customized. The three principles are:

- **Initialization**: A non-randomized cross- or multi-lingual initialization that represents a knowledge prior to bootstrap the UMT model. For instance, Lample et al. (2018a) and Artetxe et al. (2019) use an unsupervised word-translation model MUSE (Lample et al., 2018b) as initialization to promote word-to-word cross-lingual transfer. Lample et al. (2018c) use a shared jointly trained sub-word (Sennrich et al., 2016b) dictionary. On the other hand, Conneau & Lample (2019) use a pretrained cross-lingual masked language model (XLM) to initialize the unsupervised NMT model.

- **Language modeling**: Training a language model on monolingual data helps the UMT model to generate fluent texts. The neural UMT approaches (Lample et al., 2018a;c; Conneau & Lample, 2019) use denoising auto-encoder training to achieve language modeling effects in the neural model. Meanwhile, the PBSMT variant proposed by Lample et al. (2018c) uses the KenLM smoothed n-gram language models (Heafield, 2011).

- **Iterative back-translation**: Back-translation (Sennrich et al., 2016a) brings about the bridge between source and target languages by using a backward model that translates data from target to source. The (source and target) monolingual data is translated back and forth iteratively to progress the UMT model in both directions.

During training, the initialization step is conducted once, while the denoising and back-translation steps are often executed in an alternating manner.[3] It is worth noting that depending on different implementations, the parameters for backward and forward components may be separate (Lample et al., 2018a) or shared (Lample et al., 2018c; Conneau & Lample, 2019). A parameter-shared cross-lingual NMT model has the capability to translate from either source or target, while a UMT system with parameter-separate models has to maintain two models. Either way, we deem a standard UMT system to be bidirectional, i.e. it is capable of translating from either source or target language.

---

[3]The KenLM language model in PBSMT (Lample et al., 2018c) was kept fixed during the training process.

Our proposed cross-model back-translated distillation (CBD) works outside this well-established framework. It employs two UMT agents to create extra diversified data apart from what existing methods already offer, rendering it a useful add-on to the general UMT framework. Furthermore, different implementations of UMT as discussed above can be plugged into the CBD system to achieve a performance boost, even for future methods that may potentially not employ the three principles.

## 3  CROSS-MODEL BACK-TRANSLATED DISTILLATION

Let $\mathbb{X}_s$ and $\mathbb{X}_t$ be the sets of monolingual data for languages $s$ and $t$, respectively. We first train two UMT agents independently with two different parameter sets $\theta_1$ and $\theta_2$ using existing methods (Lample et al., 2018a;c; Conneau & Lample, 2019).[4] Since a UMT agent with parameter set $\theta$ is deemed bidirectional in our setup, we denote $y_t \sim P(\cdot|x_s, \theta)$ to be the translation sample from language $s$ to $t$ of input sentence $x_s$ using parameters $\theta$. In addition, for each language direction $s \to t$ (and vice versa for $t \to s$), we introduce a *supervised* MT model with parameters $\hat{\theta}$ that will be trained through our CBD. With these notations, we denote the following training loss $\mathcal{L}$ with respect to input variables $\theta_\alpha, \theta_\beta \in \{\theta_1, \theta_2\}$, the supervised parameters $\hat{\theta}$ and the learning rate $\eta$. Figure 1 also illustrates the translation flows of function $\mathcal{L}$.

$$
\begin{aligned}
\mathcal{L}(\theta_\alpha, \theta_\beta, \hat{\theta}) = &-\mathbb{E}_{z_s \sim P(\cdot|y_t, \theta_\beta), y_t \sim P(\cdot|x_s, \theta_\alpha), x_s \sim \mathbb{X}_s} \big[ \log P(y_t|x_s, \hat{\theta}) + \log P(y_t|z_s, \hat{\theta}) \big] \\
&-\mathbb{E}_{z_t \sim P(\cdot|y_s, \theta_\beta), y_s \sim P(\cdot|x_t, \theta_\alpha), x_t \sim \mathbb{X}_t} \big[ \log P(x_t|y_s, \hat{\theta}) + \log P(z_t|y_s, \hat{\theta}) \big]
\end{aligned} \tag{1}
$$

In the CBD strategy (Algorithm 1, Eq. 1), at each step, each agent $\theta \in \{\theta_1, \theta_2\}$ samples translations from the monolingual data of both languages $s$ and $t$ to acquire the *first level* of synthetic parallel data $(x_s, y_t)$ and $(x_t, y_s)$. In the *second level*, the other agent $\{\theta_1, \theta_2\} \setminus \theta$ is used to sample the translation $z_s$ of the translation $y_t$ of $x_s$ (and similarly for $z_t$ from the translation $y_s$ of $x_t$). This process is basically back-translation, but with the backward model coming from a different regime than that of the forward model. The fact that the first level agent must be different from the second level agent is crucial to achieve the desirable level of diversity in data generation. After this, we update the final model $\hat{\theta}$ using all the synthetic pairs $(x, y)$ and $(y, z)$ with maximum likelihood estimation (MLE).

---

**Algorithm 1** Cross-model Back-translated Distillation: Given monolingual data $\mathbb{X}_s$ and $\mathbb{X}_t$ of languages $s$ and $t$, return a $s \to t$ UMT model with parameters $\hat{\theta}$.

---

1: **procedure** CBD($s, t$)
2:     Train the 1st UMT agent with parameters $\theta_1$
3:     Train the 2nd UMT agent with parameters $\theta_2$
4:     Randomly initialize the parameters for MT model, $\hat{\theta}$
5:     **while** until convergence **do**
6:         $\hat{\theta} \leftarrow \hat{\theta} - \eta \nabla_{\hat{\theta}} \mathcal{L}(\theta_1, \theta_2, \hat{\theta})$
7:         $\hat{\theta} \leftarrow \hat{\theta} - \eta \nabla_{\hat{\theta}} \mathcal{L}(\theta_2, \theta_1, \hat{\theta})$
8:     **return** $\hat{\theta}$

$$X_s \dashrightarrow x_s \xrightarrow[s \to t]{\theta_\alpha} y_t \xrightarrow[t \to s]{\theta_\beta} z_s \xrightarrow[(z_s, y_t)]{(x_s, y_t)} \hat{\theta}$$

$$X_t \dashrightarrow x_t \xrightarrow[t \to s]{\theta_\alpha} y_s \xrightarrow[s \to t]{\theta_\beta} z_t \xrightarrow[(y_s, z_t)]{(y_s, x_t)} \hat{\theta}$$

Figure 1: Translation flows of loss function $\mathcal{L}$ in Eq. 1. Variable set $(\theta_\alpha, \theta_\beta)$ is replaced with $(\theta_1, \theta_2)$ and $(\theta_2, \theta_1)$ iteratively in Algorithm 1. Parallel pairs $(x_s, y_t), (z_s, y_t), (y_s, x_t)$ and $(y_s, z_t)$ are used to train the final model $\hat{\theta}$.

In this way, the model $\hat{\theta}$ first gets trained on the translated products $(x - y)$ of the UMT teachers, making it as capable as the teachers. Secondly, the model $\hat{\theta}$ is also trained on the second-level data $(y - z)$, which is slightly different from the first-level data. Thus, this mechanism provides extra data diversification to the system $\hat{\theta}$ in addition to what the UMT teachers already offer, resulting in our final model outperforming the UMT baselines (§4). However, one may argue that since $\theta_1$ and $\theta_2$ are trained in a similar fashion, $z$ will be the same as $x$, resulting in a duplicate pair. In our experiments, on the contrary, the back-translated dataset contains only around 14% duplicates across different language pairs, as shown in our analysis on data diversity in §5.2.In Appendix §8.1, we provide a more generalized version of CBD with $n (\geq 2)$ UMT agents, where we also analyze its effectiveness on the IWSLT translation tasks.

---

[4] For neural approaches, changing the random seeds would do the trick, while PBSMT methods would need to randomize the initial embeddings and/or subsample the training data.

## 4 EXPERIMENTS

We present our UMT experiments on the WMT tasks (§4.1) followed by IWSLT (§4.2).

### 4.1 WMT EXPERIMENTS

**Setup.** We use the default data setup and codebase of Conneau & Lample (2019)'s code repository[5]. Specifically, we use the News Crawl 2007-2008 datasets for English (En), French (Fr) and German (De), and News Crawl 2015 dataset for Romanian (Ro), and limit the total number of sentences per language to 5M. This results in a total 10M, 10M and 7M sentences of combined monolingual data for En-Fr, En-De and En-Ro, respectively. Note that in their paper, Lample et al. (2018c) and Conneau & Lample (2019) conducted their UMT experiments on very large datasets of 274M, 509M and 195M sentences (combination of source and target monolingual data) for En-Fr, En-De and En-Ro respectively, which are derived from News Crawl 2007-2017 datasets. Unfortunately, we were unable to reproduce these (very) large scale experiments because we encountered out-of-CPU-memory error when attempting to load the large datasets, even though our system has 128GB RAM. Instead, we rerun the NMT and PBSMT models (Lample et al., 2018c) and the pretrained XLM (Conneau & Lample, 2019) on the default setup to acquire the corresponding NMT, PBSMT and XLM baselines.

For the NMT models, we follow Lample et al. (2018c) to train the UMT agents with a parameter-shared Transformer (Vaswani et al., 2017) that has 6 layers and 512 model dimensions and a batch size of 32 sentences. We use joint Byte-Pair Encoding (BPE) (Sennrich et al., 2016b) and train fastText (Bojanowski et al., 2017) on the BPE tokens to initialize the token embeddings. For the PBSMT (Koehn et al., 2003) models, following Lample et al. (2018c), we use MUSE (Lample et al., 2018b) to generate the initial phrase table and run 4 iterations of back-translation. We subsample 500K sentences from the 5M monolingual sentences at each iteration to train the PBSMT models.[6] To ensure randomness in the PBSMT agents, we use different seeds for MUSE training and randomly subsample different sets of data during PBSMT training. For XLM (Conneau & Lample, 2019), we also follow their setup. For each agent, we first pretrain a big XLM model (6 layers, 1024 model dimensions) on the default 10M data with 4K tokens per batch. Then we initialize the encoder-decoder system of the UMT agent with the XLM and train the model for 30 epochs.[6] The NMT and XLM agents are trained with a 4-GPU setup. We use the big Transformer (Ott et al., 2018) as the final supervised model. We choose the best model based on validation loss and use beam size of 5.

Table 1: BLEU scores on WMT'14 English-French (En-Fr), WMT'16 English-German (En-De) and WMT'16 English-Romanian (En-Ro) unsupervised translation tasks.

| Method / Data | En-Fr | Fr-En | En-De | De-En | En-Ro | Ro-En |
|---|---|---|---|---|---|---|
| **Results reported from previous papers on large scale datasets** | | | | | | |
| **Data Used** | 274M | 274M | 509M | 509M | 195M | 195M |
| NMT (Lample et al., 2018c) | 25.1 | 24.2 | 17.2 | 21.0 | 21.1 | 19.4 |
| PBSMT (Lample et al., 2018c) | 27.8 | 27.2 | 17.7 | 22.6 | 21.3 | 23.0 |
| XLM (Conneau & Lample, 2019) | 33.4 | 33.3 | 26.4 | 34.3 | 33.3 | 31.8 |
| **Results from our runs on smaller datasets** | | | | | | |
| **Data Used** | 10M | 10M | 10M | 10M | 7M | 7M |
| NMT (Lample et al., 2018c) | 24.7 | 24.5 | 14.5 | 18.2 | 16.7 | 16.3 |
| + CBD | 26.5 | 25.8 | 16.6 | 20.4 | 18.3 | 17.7 |
| PBSMT (Lample et al., 2018c) | 17.1 | 16.4 | 10.9 | 13.6 | 10.5 | 11.7 |
| + CBD | 21.6 | 20.6 | 15.0 | 17.7 | 11.3 | 14.5 |
| XLM (Conneau & Lample, 2019) | 33.0 | 31.5 | 23.9 | 29.3 | 30.6 | 27.9 |
| + CBD | 35.3 | 33.0 | 26.1 | 31.5 | 32.2 | 29.1 |

---

[5]https://github.com/facebookresearch/XLM
[6]For PBSMT, Lample et al. (2018c) subsampled 5M out of 193M sentences of monolingual data. For NMT, they train for 45 epochs, which is almost 60x longer than us with data size taken into account.

**Results.** Table 1 shows the experimental results of different UMT approaches with and without CBD. First of all, with the datasets that are 30-50 times smaller than the ones used in previous papers, the baselines perform around 2 to 3 BLEU worse than the reported BLEU. As it can be seen, the CBD-enhanced model with the pretrained XLM achieves 35.3 and 33.0 BLEU on the WMT'14 En-Fr and Fr-En tasks respectively, which are 2.3 and 1.5 BLEU improvements over the baseline. It also surpasses Conneau & Lample (2019) by 1.9 BLEU, despite the fact that the SOTA was trained with 274M sentences (compared to our setup of 10M sentences). CBD also boosts the scores for XLM in En-De, De-En, En-Ro, Ro-En by around 2.0 BLEU. For the NMT systems, CBD also outperforms the baselines by 1 to 2 BLEU. More interestingly, PBSMT models are known to be deterministic, but CBD is still able to improve data diversity and provide performance boost by up to 4.0 BLEU points.

## 4.2 IWSLT EXPERIMENTS

We also demonstrate the effectiveness of CBD on relatively small datasets for IWSLT En-Fr and En-De translation tasks. The IWSLT'13 En-Fr dataset contains 200K sentences for each language. We use the IWSLT15.TED.tst2012 set for validation and the IWSLT15.TED.tst2013 set for testing. The IWSLT'14 En-De dataset contains 160K sentences for each language. We split it into 95% for training and 5% for validation, and we use IWSLT14.TED.{dev2010, dev2012, tst2010,tst1011, tst2012} for testing. For these experiments, we use the neural UMT method (Lample et al., 2018c) with a Transformer of 5 layers and 512 model dimensions, and trained using only 1 GPU.

From the results in Table 2, we can see that CBD improves the performance in all the four tasks by 2-3 BLEU compared to the NMT baseline of Lample et al. (2018c).

Table 2: BLEU scores on the unsupervised IWSLT'13 English-French (En-Fr) and IWSLT'14 English-German (En-De) tasks.

| Method / Data | En-Fr | Fr-En | En-De | De-En |
|---|---|---|---|---|
| NMT (Lample et al., 2018c) | 29.6 | 30.7 | 15.8 | 19.1 |
| + CBD | 31.8 | 31.9 | 18.5 | 21.6 |

## 5 UNDERSTANDING CBD

### 5.1 CROSS-MODEL BACK-TRANSLATION IS KEY

As mentioned, crucial to our strategy's success is the cross-model back-translation, where the agent operating at the first level must be different from the one in the second level. To verify this, we compare CBD with similar variants that do not employ the cross-model element in the WMT tasks. We refer to these variants commonly as *back-translation distillation* (BD). The first variant *BD (1st level,1 agent)* has only 1 UMT agent that translates the monolingual data only once and uses these synthetic parallel pairs to distill the model $\hat{\theta}$. The second variant *BD (1st level, 2 agents)* employs 2 UMT agents, similar to CBD, to produce 2 sets of synthetic parallel data from the monolingual data and uses both of them for distillation. Finally, the third variant *BD (2nd level, 2 agents)* uses 2 UMT agents to sample translations from the monolingual data in forward and backward directions using the same respective agents. In other words, BD (2nd level, 2 agents) follows similar procedures in Algorithm 1, except that it optimizes the following loss with $\theta_\alpha$ being alternated between $\theta_1$ and $\theta_2$:

$$\overline{\mathcal{L}}(\theta_\alpha, \hat{\theta}) = -\mathbb{E}_{z_s \sim P(\cdot|y_t, \theta_\alpha), y_t \sim P(\cdot|x_s, \theta_\alpha), x_s \sim \mathbb{X}_s} \left[ \log P(y_t|x_s, \hat{\theta}) + \log P(y_t|z_s, \hat{\theta}) \right]$$
$$- \mathbb{E}_{z_t \sim P(\cdot|y_s, \theta_\alpha), y_s \sim P(\cdot|x_t, \theta_\alpha), x_t \sim \mathbb{X}_t} \left[ \log P(x_t|y_s, \hat{\theta}) + \log P(z_t|y_s, \hat{\theta}) \right]$$

(2)

From the comparison in Table 3, we see that none of the BD variants noticeably improves the performance across the language pairs, while CBD provides consistent gains of 1.0-2.0 BLEU. In particular, the BD (1st level, 1 agent) variant fails to improve as the distilled model is trained on the same synthetic data that the UMT agent is already trained on. The variant BD (1st level, 2 agents) is in fact similar in sprit to (Nguyen et al., 2020), which improves supervised and semi-supervised MT. However, it fails to do so in the unsupervised setup, due to the lack of supervised agents. The variant BD (2nd level, 2 agents) also fails because the 2nd level synthetic data is already optimized

during iterative back-translation training of the UMT agents, leaving the distilled model with no extra information to exploit. On the other hand, cross-model back-translation enables CBD to translate the second-level data by an agent other than the first-level agent. In this strategy, the second agent produces targets that the first agent is not aware of, while the second agent receives as input the sources that are foreign to it. This process creates corrupted but new information, which the supervised MT model can leverage to improve the overall MT performance through more data diversity.

Table 3: BLEU comparison of CBD vs. no cross-model variants in the WMT'14 English-French (En-Fr), WMT'16 English-German (En-De) and English-Romanian (En-Ro) tasks.

| Method | En-Fr | Fr-En | En-De | De-En | En-Ro | Ro-En |
|---|---|---|---|---|---|---|
| NMT Baseline (Lample et al., 2018c) | 24.7 | 24.5 | 14.5 | 18.2 | 16.7 | 16.3 |
| BD (1st level, 1 agent) | 24.5 | 24.4 | 13.8 | 17.6 | 16.0 | 15.9 |
| BD (1st level, 2 agents) | 24.6 | 24.6 | 14.0 | 17.9 | 16.5 | 16.1 |
| BD (2nd level, 2 agents) | 24.7 | 24.7 | 14.4 | 18.0 | 16.7 | 16.4 |
| CBD (2nd level, 2 agents) | **26.5** | **25.8** | **16.6** | **20.4** | **18.3** | **17.7** |

## 5.2 CBD PRODUCES DIVERSE DATA

Having argued that cross-model back-translation creates extra information for the supervised MT model to leverage on, we hypothesize that such extra information can be measurable by the diversity of the generated data. To measure this, we compute the *reconstruction BLEU* and compare the scores for BD and CBD (both 2nd level, 2 agents) in the WMT En-Fr, En-De and En-Ro tasks. The scores are obtained by using the first agent to translate the available monolingual data in language $s$ to $t$ and then the second agent to translate those translations back to language $s$. After that, a BLEU score is measured by comparing the reconstructed text with the original text. In BD, the first and second agents are identical, while they are distinct for CBD. From the results in Table 4, we observe that the reconstruction BLEU scores of CBD are more than 10 points lower than those of BD, indicating that the newly generated data by CBD are more diverse and different from the original data.

Table 4: Reconstruction BLEU scores of BD and CBD in different languages for the WMT unsupervised translation tasks. Lower BLEU means more diverse.

| Method | En-Fr-En | Fr-En-Fr | En-De-En | De-En-De | En-Ro-En | Ro-En-Ro |
|---|---|---|---|---|---|---|
| BD | 76.5 | 72.7 | 75.1 | 63.3 | 73.6 | 71.2 |
| CBD | 63.1 | 59.5 | 60.7 | 50.6 | 61.0 | 56.8 |

In Table 5, we further report the ratio of duplicate source-target pairs to the amount of synthetic parallel data created by CBD. We sample 30M synthetic parallel data using the CBD strategy and examine the amount of duplicate pairs for the WMT En-Fr, En-De and En-Ro tasks. We can notice that across the language pairs, only around 14% of the parallel data are duplicates. Given that only about 5M (3.5M for En-Ro) sentences are *real* sentences and the remaining 25M sentences are synthetic, this amount of duplicates is surprisingly low. This fact also explains why CBD is able to exploit extra information better than any standard UMT to improve the performance.

Table 5: Comparison between the amount of real data, generated data by CBD and the duplicates per language pair for the WMT'14 En-Fr, WMT'16 En-De and En-Ro unsupervised MT tasks.

| Method | En-Fr | En-De | En-Ro |
|---|---|---|---|
| Real data | 5M | 5M | 3.5M |
| Generated data | 30M | 30M | 29M |
| Duplicate pairs | 4.4M (14.5%) | 3.9M(13%) | 3.9M (13.4%) |

## 5.3 COMPARISON WITH ENSEMBLES OF MODELS AND ENSEMBLE KNOWLEDGE DISTILLATION

Since CBD utilizes outputs from two UMT agents for supervised distillation, it is interesting to see how it performs compared to an ensemble of UMT models and ensemble knowledge distillation (Freitag et al., 2017). To perform ensembling, we average the probabilities of the two UMT agents at each decoding step. For ensemble distillation, we generate synthetic parallel data from an ensemble of UMT agents, which is then used to train the supervised model.

Table 6: BLEU comparison of CBD vs. an ensemble of UMT agents and ensemble knowledge distillation (Freitag et al., 2017) on WMT'14 En-Fr, WMT'16 En-De and En-Ro translation tasks.

| Method | En-Fr | Fr-En | En-De | De-En | En-Ro | Ro-En |
|---|---|---|---|---|---|---|
| NMT Baseline (Lample et al., 2018c) | 24.7 | 24.5 | 14.5 | 18.2 | 16.7 | 16.3 |
| Ensemble of 2 NMT agents | 25.2 | 24.8 | 15.3 | 19.1 | 17.7 | 17.1 |
| Ensemble distillation | 17.3 | 20.0 | 3.5 | 3.7 | 1.2 | 1.1 |
| CBD with NMT | **26.5** | **25.8** | **16.6** | **20.4** | **18.3** | **17.7** |

From the results on the WMT translation tasks in Table 6, we observe that ensembles of models improve the performance only by 0.5-1.0 BLEU, while CBD provides larger gains (1.0-2.0 BLEU) across all the tasks. These results demonstrate that CBD is capable of leveraging the potentials of multiple UMT agents better than how an ensemble of agents does. This is in contrast to data diversification (Nguyen et al., 2020), which is shown to mimic and perform similarly to model ensembling. More importantly, during inference, an ensemble of models requires more memory and computations (twice in this case) to store and execute multiple models. In contrast, CBD can throw away the UMT teacher agents after training and needs only one single model for inference. Meanwhile, ensemble knowledge distillation (Freitag et al., 2017), which performs well with supervised agents, performs poorly in unsupervised MT tasks. The reason could be that the UMT agents may not be suitable for the method originally intended for supervised learning. Further inspection in Appendix §8.2 suggests that many samples in the ensemble translations contain incomprehensible repetitions.

## 5.4 COMPARISON WITH OTHER POTENTIAL ALTERNATIVES

In this section, we compare CBD with other alternatives in the text generation literature that also attempt to increase diversity. While many of these methods are generic, we adopt them in the UMT framework and compare their performance with our CBD technique in the WMT En-Fr, Fr-en, En-De, and De-en tasks, taking the XLM (Conneau & Lample, 2019) as the base model.

One major group of alternatives is *sampling* based methods, where the model samples translations following multinomial distributions during iterative back-translation. Specifically, we compare the CBD with (*i*) sampling with temperature 0.3 (Edunov et al., 2018; Fan et al., 2018), (*ii*) top-k sampling (Radford et al., 2019), and (*iii*) nucleus or top-p sampling (Holtzman et al., 2020). Plus, we compare CBD with *target noising*, where we add random noises to the translations of the UMT model during iterative back-translation. Finally, multi-agent dual learning (Wang et al., 2019) is also considered as another alternative, where multiple unsupervised agents are used train the end supervised model.

The results are reported in Table 7. We can see that while the sampling based methods indeed increase the diversity significantly, they do not improve the performance as much as CBD does. The reason could be that the extra data generated by (stochastic) sampling are noisy and their quality is not as good as deterministic predictions from the two UMT agents via cross-model back-translation. On the other hand, target noising does not provide a consistent improvement while multi-agent dual learning achieves less impressive gains compared to CBD.

## 6 RELATED WORK

The first step towards utilizing the vast monolingual data to boost MT quality is through semi-supervised training. Back-translation (Sennrich et al., 2016a; Edunov et al., 2018) is an effective approach to exploit target-side monolingual data. Dual learning (He et al., 2016; Wang et al., 2019), meanwhile, trains backward and forward models concurrently and intertwines them together. Recently, Zheng et al. (2020) proposed a variational method to couple the translation and language models

Table 7: Comparison with other alternatives on the WMT En-Fr, Fr-En, En-De and De-En, with XLM as the base model.

| WMT | En-Fr | Fr-En | En-De | De-En |
|---|---|---|---|---|
| XLM Baseline (Conneau & Lample, 2019) | 33.0 | 31.5 | 23.9 | 29.3 |
| Sampling with temperature 0.3 | 33.5 | 32.2 | 24.3 | 30.2 |
| Top-k sampling | 33.18 | 32.26 | 24.0 | 29.9 |
| Nucleus (top-p) sampling | | Diverge | | |
| Target noising | 32.8 | 30.7 | 24.0 | 29.6 |
| Multi-agent dual learning | 33.5 | 31.7 | 24.6 | 29.9 |
| CBD | 35.3 | 33.0 | 26.1 | 31.5 |

through a shared latent space. There have also been attempts in solving low-resource translation problems with limited parallel data (Gu et al., 2018; Irvine & Callison-Burch, 2014; Guzmán et al., 2019). In the realm of SMT, cross-lingual dictionaries have been used to reduce parallel data reliance (Irvine & Callison-Burch, 2016; Klementiev et al., 2012).

In recent years, unsupervised word-translation via cross-lingual word embedding has seen a huge success (Lample et al., 2018b; Artetxe et al., 2017; 2018a). This opened the door for UMT methods that employ the three principles described in §2. Lample et al. (2018a) and Artetxe et al. (2018c) were among the first of this kind, who use denoising autodecoder for language modeling and iterative back-translation. Lample et al. (2018a) use MUSE (Lample et al., 2018b) as the initialization to bootstrap the model, while Artetxe et al. (2018c) use cross-lingual word embeddings (Artetxe et al., 2017). Lample et al. (2018c) later suggested the use of BPE (Sennrich et al., 2016b) and fastText (Bojanowski et al., 2017) to initialize the model and the parameters sharing. Pretrained language models (Devlin et al., 2019) are then used to initialize the entire network (Conneau & Lample, 2019). Song et al. (2019) proposed to pretrain an encoder-decoder model while Artetxe et al. (2018b) suggested a combination of PBSMT and NMT with subword information.[7] Plus, pretraining BART (Lewis et al., 2020) on multi-lingual corpora improves the initialization process (Liu et al., 2020).

Our proposed CBD works outside the three-principle UMT framework and is considered as an add-on to any underlying UMT system. There exist some relevant approaches to CBD. First, it is similar to Nguyen et al. (2020), which generates a diverse set of data from multiple *supervised* MT agents. Despite being effective in supervised and semi-supervised settings, a direct implementation of it in UMT underperforms due to lack of supervised signals (§5.1). In order to successfully exploit unsupervised agents, CBD requires cross-model back-translation which is the key to its effectiveness. Second, CBD can be viewed as an augmentation technique (Fadaee et al., 2017; Wang et al., 2018). Although the denoising autoencoding built in a typical UMT system also performs augmentation, the noising process is rather naive, while CBD augments data by well-trained agents. Third, CBD is related to ensemble knowledge distillation (Kim & Rush, 2016; Freitag et al., 2017). These methods use multiple models to perform ensemble inference to generate one-way synthetic targets from the original source data, which are then used to distill the final model. However, these distillation schemes only apply to supervised settings and the agents are used to jointly produce high quality translations rather than diverse candidates. Similar to our method, multi-agent dual learning (Wang et al., 2019) also uses multiple models in both forward and backward directions, but to minimize the reconstruction losses instead of to generate diverse synthetic data.

## 7 CONCLUSION

We have proposed cross-model back-translated distillation (CBD) - a method that works outside the three existing principles for unsupervised MT and is applicable to any UMT methods. CBD improves the performance of various underlying UMT approaches in the WMT'14 English-French, WMT'16 English-German and English-Romanian translation tasks by 1.0-2.0 BLEU. It also outperforms the baselines in the IWSLT'14 German-English and IWSLT'13 English-French tasks by up to 3.0 BLEU. Our analysis shows that CBD embraces data diversity and extracts more model-specific intrinsic information than what an ensemble of models would do.

---

[7]Since Artetxe et al. (2018b) did not provide the code, we were unable to apply CBD to their work.

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

---

**Algorithm 2** Generalized Cross-model Back-translated Distillation (GCBD): Given monolingual data $\mathbb{X}_s$ and $\mathbb{X}_t$ of languages $s$ and $t$, and hyper-parameter $n$, return a $s \rightarrow t$ UMT model with parameters $\hat{\theta}$.

---

1: **procedure** GCBD$(n, s, t)$
2:     **for** $i \in 1, ..., n$ **do**
3:         Train UMT agent with parameters $\theta_i$
4:     Randomly initialize the parameters for the MT model, $\hat{\theta}$
5:     **while** until convergence **do**
6:         **for** $i \in 1, ..., n$ **do**
7:             **for** $j \in 1, ..., n$ where $j \neq i$ **do**
8:                 $\hat{\theta} \leftarrow \hat{\theta} - \eta \nabla_{\hat{\theta}} \mathcal{L}(\theta_i, \theta_j, \hat{\theta})$
9:     **return** $\hat{\theta}$

---

# 8 APPENDIX

In the following supplementary material, we provide the generalized version of our method cross-model back-translated distillation, or GCBD, and measure its effectiveness in the IWSLT English-German, German-English, English-French and French-English unsupervised tasks (§8.1). In addition, we investigate why ensemble knowledge distillation (Freitag et al., 2017), which boosts the performance in a supervised setup, fails to do so in an unsupervised setup where we replace the supervised agents used in the method with the UMT agents (§8.2). Lastly, We also show that our method is uneffected by the translationese effect (Edunov et al., 2020) (§8.3).

## 8.1 GENERALIZED VERSION

In this section, we describe a generalized version of our CBD, which involves multiple UMT agents instead of just two. Then, we test this method in the IWSLT experiments to demonstrate its effectiveness and characteristics. Specifically, in addition to the input monolingual data $\mathbb{X}_s$ and $\mathbb{X}_t$ of languages $s$ and $t$ and the supervised model $\hat{\theta}$, we introduce another hyper-parameter $n$ to indicate the number of unsupervised agents used to perform cross-model back-translation. The generalized cross-model back-translated distillation (GCBD) strategy is presented in Algorithm 2. In this method, instead of training only two agents, the method trains a set of $n$ UMT agents $\Theta = \{\theta_1, ..., \theta_n\}$. During training, we iteratively select two orderly distinct agents $\theta_i$ and $\theta_j$ from $\Theta$ and use them to perform cross-model back-translation and train the model $\hat{\theta}$.

To evaluate GCBD in comparison with CBD, we conduct experiments with the IWSLT'13 English-French (En-Fr) and IWSLT'14 English-German (En-De) tasks. The setup for these experiments are identical to the experiment in §4.2, except that we vary the hyper-parameter $n = (2, 3, 4)$ to determine the optimal number of agents. The results are reported in Table 8. As it can be seen, increasing the number of agents $n$ to 3 adds an additional $0.4 - 1.0$ BLEU improvement compared to the standard CBD. Moreover, using 4 UMT agents does not improve the performance over using just 3 UMT, despite that this setup still outperforms the standard CBD. The results indicate that increasing the system complexity further is not always optimal and diminishing return is observed as we add more agents to the system.

Table 8: BLEU scores on the unsupervised IWSLT'13 English-French (En-Fr) and IWSLT'14 English-German (En-De) tasks with varying number of agents $n$ of GCBD.

| Method / Data | En-Fr | Fr-En | En-De | De-En |
|---|---|---|---|---|
| NMT (Lample et al., 2018c) (baseline) | 29.6 | 30.7 | 15.8 | 19.1 |
| + GCBD ($n = 2$) or CBD | 31.8 | 31.9 | 18.5 | 21.6 |
| + GCBD ($n = 3$) | **32.8** | **32.2** | **19.1** | **22.2** |
| + GCBD ($n = 4$) | 32.3 | 32.1 | 19.1 | 22.1 |

## 8.2 Analysis of degeneration in ensemble knowledge distillation

Ensemble knowledge distillation (Freitag et al., 2017) has been used to enhance supervised machine translation. It uses multiple strong (supervised) teachers to generate synthetic parallel data from both sides of the parallel corpora by averaging the decoding probabilities of the teachers at each step. The synthetic data are then used to train the student model. Having seen its effectiveness in the supervised setup, we apply this same tactic to unsupervised MT tasks by replacing the supervised teachers with unsupervised MT agents. However, the method surprisingly causes drastic performance drop in the WMT'14 En-Fr, WMT'16 En-De and En-Ro unsupervised MT tasks.

By manual inspection, we found that many instances of the synthetic data are incomprehensible and contain repetitions, which is a degeneration behavior. We then quantitatively measure the percentage of sentences in the synthetic data containing tri-gram repetitions by counting the number of sentences where a word/sub-word is repeated at least three consecutive times. As reported in Table 9, from 30% to 86% of the synthetic data generated by the ensemble knowledge distillation method are incomprehensible and contain repetitions. Relative to the performance of CBD, the performance drop in ensemble distillation is also more dramatic for language pairs with higher percentage of degeneration (En-Ro and En-De). This explains why the downstream student model fails to learn from these corrupted data. The results indicate that UMT agents are unable to jointly translate through ensembling strategy the monolingual data that they were trained on. This phenomenon may require further research to be fully understood.

On the other hand, with less than 0.1% tri-gram repetitions, CBD generates little to no repetitions, which partly explains why it is able to improve the performance.

Table 9: Percentage of tri-gram repetitions in the synthetic data generated by ensemble knowledge distillation (Freitag et al., 2017), compared to those created by CBD; and the respective test BLEU scores in WMT'14 En-Fr, WMT'16 En-De and En-Ro unsupervised tasks.

| Method | En-Fr | Fr-En | En-De | De-En | En-Ro | Ro-En |
|---|---|---|---|---|---|---|
| **% tri-gram repetition** | | | | | | |
| Ensemble distillation | 30.3% | 34% | 73% | 76% | 43% | 86% |
| CBD (2nd level, $n = 2$) | $10^{-3}$% | $10^{-2}$% | $10^{-2}$% | $10^{-1}$% | $10^{-2}$% | $10^{-2}$% |
| **BLEU on test set** | | | | | | |
| Ensemble distillation | 17.3 | 20.0 | 3.5 | 3.7 | 1.2 | 1.1 |
| CBD (2nd level, $n = 2$) | 26.5 | 25.8 | 16.6 | 20.4 | 18.3 | 17.7 |

## 8.3 The translationese effect

It can be seen that our cross-model back-translation method is indeed a modified version of back-translation (Sennrich et al., 2016a). Therefore, it is necessary to test if this method suffer from the *translationese effect* (Edunov et al., 2020). As pointed out in their work, back-translation only shows performance gains with translationese source sentences but does not improve when the sentences are natural text.[8] Nguyen et al. (2020) shows that the translationese effect only exhibits in semi-supervised setup, where there are both parallel and monolingual data. However, while they show that their supervised back-translation technique is not impacted by the translationese effect, they left out the question whether unsupervised counterparts are affected.

Therefore, we test our unsupervised CBD method against the translationese effect by conducting the same experiment. More precisely, we compare the BLEU scores of our method versus the XLM baseline (Conneau & Lample, 2019) in the WMT'14 English-German test sets in the three setups devised by Edunov et al. (2020):

- Natural source $\rightarrow$ translationese target ($X \rightarrow Y^*$).
- Translationese source $\rightarrow$ natural target ($X^* \rightarrow Y$)
- Translationese of translationese of source to translationese of target ($X^{**} \rightarrow Y^*$).

---

[8]Translationese is human translation of a natural text by a professional translator. Translationese tends to be simpler, more grammatically correct, but lacks contextual sentiments and fidelity.

Table 10 shows that our method outperforms the baseline significantly in the natural source $\rightarrow$ translationese target scenario ($X \rightarrow Y^*$), while it may not improve the translationese source scenario ($X^* \rightarrow Y$) considerably. The results demonstrate that our method behaves differently than what the translationese effect indicates. More importantly, the translations of the natural source sentences are improved, which indicates the practical usefulness of our method. Furthermore, in combination with the findings in Nguyen et al. (2020), the experiment shows that the translationese effect may only exhibit in the semi-supervised setup, but not in supervised or unsupervised setups.

Table 10: BLEU scores of our method and the baseline (Conneau & Lample, 2019) on the translationese effect (Edunov et al., 2020), in the WMT'14 English-German setup.

| **WMT'14 En-De** | $X \rightarrow Y^*$ | $X^* \rightarrow Y$ | $X^{**} \rightarrow Y^*$ |
|---|---|---|---|
| UMT Baseline (XLM) | 18.63 | 18.01 | 25.59 |
| Our method | 20.79 | 18.24 | 27.85 |

