# OpenReview forum: "Cross-model Back-translated Distillation for Unsupervised Machine Translation"
_ICLR.cc/2021/Conference — Reject_

### Official Review · AnonReviewer4 · 2020-10-27
**This paper introduce a novel data enhancement approach for unsupervised neural machine translation.**

**Rating:** 5
**Confidence:** 5

**Review:**

In this paper, two unsupervised agents are utilized at cross-model by using the dual nature of the unsupervised machine translation model, in which forward translation of agent_1 is combined with the backward translation of agent_2, more synthetic translation pairs are obtained to train a new supervised machine translation model. The result is improved on multiple unsupervised machine translation, and this paper claims that more diversity is brought to the synthetic data, so a better translation model can be trained. This paper uses a reconstruction BLEU or BT BLEU [1] metric to compare the effect of the inside-model with that of cross-model, and finds that cross model translation has a lower back-translation effect, which shows that the diversity is enhanced. Furthermore, CBD is compared with the ensemble method and achieves better performance. The proposed method is quite simple yet effective, but it is also a kind of data enhancement.

In addition to these contributions, the paper also has some shortcomings

1. The evidence in this paper can not support the claim that the current performance bottleneck of UMT is due to the lack of diversity: the performance upper limit of UMT is still due to the lack of clear supervision signal, which limits the further performance growth. Because the training of CBD is divided into two stages, the diversity of the second stage only brings more training data to enhance the supervised machine translation model, rather than unsupervised machine translation effect.

2. Source of promotion: the second stage of CBD method adopts (x_s, y_t), (z_s, y_t), (y_s, x_t), (y_s, z_t) synthetic translation pairs, it is not clear how much performance growth comes from increased data and how much growth comes from the new model implementation (ott et al., 2018). It is not appropriate to attribute all contributions to the diversity brought by CBD. I suggest that the author should use (y_s, x_t) data to train based on the (ott et al., 2018) model, and report the effect comparison (In my experiments, the second stage model implemented with fairseq trained only on (y_s, x_t) surpass both agents trained with XLM due to more efficient implementation in fairseq).

3. Unfair comparison with the enable distillation: authors need to compare CBD with the model trained on the synthetic data (y_s, x_t) of the ensemble of agent_1 and agent_ 2. In the training data (x_s, y_t), (z_s, y_t), (y_s, x_t), (y_s, z_t) for the second stage of CDB,  x_t, golden language sequences as translation target is stronger than synthetic language sequences (silver) as target. Therefore, It is necessary to report the real result of ensembled distillation. The current results are very unreliable. In addition, it is necessary to compare the training time of the CBD method and ensemble distillation training (including the decoding process after the first stage of training) to show the efficiency of CBD.

4. The non-golden language sequence as a translation target is called pseudo-NMT (PNMT). The author adopts a variety of model structures, which is slightly redundant. They can directly add the synthetic data decoded by cross model to continually train the original XLM model with a supervised translation objective (which is naturally supported in XLM from my experience), and report the effect comparison between them.

5. The essence of the CDB approach is a process of self-supervised training, so it is necessary to compare self-training/tri-training introduced in [2].


In general, the CBD method in this paper is a simple and effective data enhancement method to improve the performance of the model. However, due to the lack of many important details of the implementation, despite the promotion, the source of promotion is unknown. In addition, the unreasonable comparison of the baseline models deepens my concern about the real promotion of this CBD method.

[1] Li, Zuchao, et al. "Reference Language based Unsupervised Neural Machine Translation." arXiv preprint arXiv:2004.02127 (2020).

[2] Sun, Haipeng, et al. "Self-Training for Unsupervised Neural Machine Translation in Unbalanced Training Data Scenarios." arXiv preprint arXiv:2004.04507 (2020).

---

### Official Review · AnonReviewer3 · 2020-10-28
**Good paper, simple approach, thorough experiments**

**Rating:** 7
**Confidence:** 5

**Review:**

Summary: The paper proposes an additional stage of training for unsupervised NMT models utilizing synthetic data generated from multiple independently trained models. The generated synthetic data uses two stages of back-translation, with different models, in order to "diversify" the set of training data used for fine-tuning the models. This is similar to the approach in [1], but uses an additional stage of back-translation with a different model. The authors add this additional stage of training to unsupervised NMT models using different pipelines (PB unsupervised MT, Neural Unsupervised MT, XLM) and show that their approach improves all of these approaches by 1.5-2 Bleu on WMT En-Fr, De-En and En-Ro.

Strengths:
1. The paper is well written, the approach is simple and seems to improve quality by significant amounts in a variety of experimental settings.
2. The authors do a great job of comparing against several relevant approaches (sampling during back-translation, ensembling, multi-agent dual learning). The paper compares against most of the relevant approaches I could think of while reading the paper.

Weaknesses / Questions for authors:
1. As with any NMT model trained with synthetic data, it would be better to report results on source and target original splits of the test data to provide a clearer evaluation [2,3]. Also clarify the Bleu scripts, tokenization and other post-processing used for evaluation.
2. The datasets used for experimentation are much smaller than the ones used for the baseline unsupervised-NMT approaches. It would be great to report results in the original training conditions (this is not a major limitation however, since the proposed approach seems to improve over baselines trained with more data).
3. Did the authors try any experiments with unsupervised models utilizing parallel data in unrelated languages, similar to [4,5] or in real low resource settings [6]? These are more practical conditions for unsupervised MT in true low resource languages.

Recommendation: Overall, this is a good paper and I would recommend acceptance. While I would have also liked to see experiments in more realistic low-resource settings, the current paper does a good enough job of evaluating the approach in standard unsupervised NMT settings on related high resource languages.

References:
[1] Data Diversification: A Simple Strategy For Neural Machine Translation, Nguyen et al.
[2] APE at Scale and its Implications on MT Evaluation Biases, Freitag et al.
[3] On The Evaluation of Machine Translation Systems Trained With Back-Translation, Edunov et al.
[4] Multilingual Denoising Pre-training for Neural Machine Translation, Liu et al.
[5] Leveraging Monolingual Data with Self-Supervision for Multilingual Neural Machine Translation, Siddhant et al.
[6] When Does Unsupervised Machine Translation Work?, Marchisio et al.

---

### Official Review · AnonReviewer1 · 2020-10-28
**Very thorough exploration of a simple, flexible idea. Unclear whether the data augmentation still matters when more monolingual data is available.**

**Rating:** 7
**Confidence:** 4

**Review:**

This paper describes a method to enhance unsupervised machine translation through data augmentation. The idea is pretty straight-forward, if not altogether intuitive, you begin by training two bidirectional (i.e.: they can translate source to target and target to source) unsupervised MT systems A and B. The tested scenarios always have A and B be identical architectures trained with different initializations. They then produce synthetic source-target pairs by first having A (source->target) translate the provided source sentence x to y, and then having B (target->source) translate y back to z. They then train supervised MT on both x,y and z,y. The same procedure can be repeated with source and target reversed. The authors show substantial (1-2) BLEU improvements with 3 different UMT systems in 5 low-data scenarios (En-Fr, Fr-En, En-De, En-Ro, Ro-En), all subsampled to 5M monolingual sentences for each language. In En-Fr and Fr-En and En-De, they are able to match reported XLM results from Conneau and Lample 2019, despite using much less data.

This simple idea is explored extremely thoroughly. The paper reads more like a journal paper that has undergone several stages of review than a conference paper. The authors make connections to and compare against a number of relevant ensembling strategies (to account for two systems being used) and back-translation-diversification strategies (to account for multiple sources being produced for the same target), and consistently show that only their specific recipe leads to the same levels of improvement. The authors really leave no stone unturned.

The biggest knock against this paper is the relatively small data scenario. Having two UMT systems allows them to provide two source sentences (one original and one synthetic) for each target sentence (always synthetic), but how important is this when we have 25x more original source sentences? I can imagine arguments for why the high-data UMT scenario is unrealistic (many monolingual sentences implies the likely presence of parallel data), but those arguments aren’t presented in the paper. It would be greatly strengthened by a full-data experiment for even just one or two of the language pairs.

Beyond that, I have few concerns. The paper is clear, easy to follow, and as I said, very thorough. But I’ll do my best to make some constructive criticisms:

I think the Related Work section feels a little superfluous after all of the comparisons made to related work in the Background and in the Experiments. I think I would like to have seen more discussion of the highlighly related work in sections 5.3 and 5.4. In particular, a more detailed discussion of this method’s relation to multi-agent dual learning would be worth giving up parts of Related work that are already mentioned in Background (like pre-neural statistical unsupervised MT).

It would be useful to specify how BLEU is calculated, to help readers understand just how useful the cross-paper BLEU comparisons in Table 1 are.

---

### Official Review · AnonReviewer2 · 2020-10-30
**Strong results but a few unclear parts in the paper**

**Rating:** 6
**Confidence:** 3

**Review:**

This paper introduces a new component to the unsupervised machine translation framework called cross-model back-translated distillation. The proposed approach is applicable to the other unsupervised methods. Experimental results in several translation tasks show that the proposed approach improves the translation accuracy of the standard unsupervised machine translation models, outperforming the cross-lingual masked language model.

- The analyses are interesting to understand the proposed approach. Table 4 reports the diverse of synthetic data, but what about the quality as parallel data? Can you use parallel data instead and conduct the same analyses so that you can use BLEU score as an evaluation metric?
- Is it possible to apply the proposed approach to supervised NMT training, by creating BT data from monolingual data?
- Table 1 reports that the results from your experiments show that the equivalent/better performance agains the existing models with much fewer data. What about scaling up the monolingual data size 5x/10x more? Will the performance be improved better and better?
- "the translated products (x-y) of the UMT teachers." at p.6.  What does this "x-y" mean?

Typo:
p.3 5.2.In Appendix -> 5.2. In Appendix

---

### Comment · Area_Chair1 · 2020-11-21
**The discussion stage is open**

Dear Reviewers:

Thanks for your insightful reviews! Now the discussion stage is open and the authors have posted their responses. We will appreciate that the following things-to-do can be done by Tues, Nov 24.

1 Acknowledge explicitly that you have read the responses.

2 Modify your review if necessary.

3 Communicate with the authors/reviewers/AC by adding/responding to the comments if necessary.

Thanks a lot!

---

### Author Response · Authors · 2021-03-23
**Discussions with Reviewer #4**

We disclose and make public the responses and exchanges between us and reviewer #4 (R4).

----[1] 1st Author Response to R4's initial review ------
1. We humbly disagree with you. Our objective is to build an effective unsupervised MT system. As long as no parallel data is used, it is fine if the model is trained with a so-called “supervised” loss function. After all, almost all NMT systems (unsupervised or supervised) are trained with cross-entropy loss functions provided by (synthetic) parallel data. The evidence shows that if we create more diversity in the synthetic parallel data, we can improve UMT performance. The empirical evidence in Section 5.1, 5.2 and 5.3 shows various alternative attempts (which use supervised loss) fail to diversify the data further and thus fail to improve the UMT performance considerably. This may not theoretically or formally prove that the lack of diversity is the bottleneck. And we haven’t found any relevant theory in the literature about diversity in MT either. But the results show practical and empirical usefulness of our method and real-life adoption potential. We believe this is a worthwhile scientific discovery. We, however, do not claim that the lack of diversity is the ONLY bottleneck, since there may be other issues of UMT that can be improved.

2. Regarding the source of promotion, we believe you unfortunately missed the key information in Section 5.1, which exactly addresses your concerns. Section 5.1 and Table 3 show clearly the effectiveness of the second stage (cross-model back-translation) and address your concern about Ott et al., 2018. All the BD variants in Table 3 were trained using the new implementation (Ott et al., 2018). The BD (1st level, 1agent) exactly “use (y_s, x_t) data to train based on the (Ott et al., 2018) model”. CBD outperforms BD by a large margin, despite the fact that both of them use the implementation of Ott et al., 2018. So the performance gain is not attributed to the better implementation.

3. In Section 5.3, we have stated “for ensemble distillation, we generate synthetic parallel data from an ensemble of UMT agents”, which is exactly what you are asking “to have a model trained on the synthetic data (y_s, x_t) of the ensemble of agent_1 and agent_2”. More specifically, we combined both agent_1 and agent_2 to perform ensemble translations of monolingual data x_t to get y_s, and then use the pair (y_s, x_t) to train the final model. Therefore, our experiments in Section 5.3 exactly address your concern about ensemble distillation. We hope this clarifies your doubt. The decoding process of CBD is twice as long as the decoding process of ensemble distillation. But both decoding processes are minimal compared to the time required for the first stage training (about 1/4 the training time). Overall, CBD is about 16% longer than ensemble distillation. More importantly, ensemble distillation performs much worse than the baseline (NMT).

4. Good question. We tried to add such synthetic data to the currently being trained UMT model in our initial experiments, like you suggested, but we got worse results. The reason is that the current UMT model is already mature and the learning rate at that time has been reduced substantially. This prevents the model from picking up new knowledge from the new diverse data. If we keep the learning rate higher, we encounter catastrophic forgetting, which also reduces the performance. These results motivated us to use a fresh model and train from scratch. Otherwise, we wouldn’t have to resort to a more resource-consuming tactic in this paper.

The mentioned self-training approach is specifically targeted at extreme unbalanced data scenarios only. In our setup, however, the data is generally balanced. So we do not think a comparison is really needed. Despite that, we conducted an experiment with self-training [2]. And the results below show that self-training does not produce much gains when the data is generally balanced.

|Model	|  En-Fr |Fr-En|
|--|--|--|
|Baseline | 33.0| 31.5|
|Self-training[2] |33.5| 32.1|
| Our method |35.3| 33.0|

----[2] R4's response to [1] ------

However, I still find the lack of enough supports for your arguments for the lack of diversity, and there is confusion in this version.

The reported worse results from adding synthetic data to the UMT model (XLM implementation) just to confirm my guess that better model implementation leads to better performance rather than diversity. Otherwise, the added data will bring further improvement. If simply change an implementation framework, one that works and one that doesn't, it doesn't comply with the claim. (Continuing training with the new synthetic data does not require retraining, nor does it require continuing training with BT alone, and the MT steps can also be used with this pseudo parallel data (bt_steps and mt_steps in XLM). Therefore, it is still unclear where the benefit of the proposed approach comes from.

---

> ### Author Response · Authors · 2021-03-23
> **Discussions with Reviewer #4 (continued)**
>
> ----[3] Author's response R4's response [2] ------
>
> Firstly, Section 5.1 shows that the baseline (with XLM implementation) and BD (with Ott et al., 2018’s implementation) are trained on the same set of synthetic data, and they perform relatively the same, which shows that there is no significant performance boost with Ott et al., 2018’s implementation.
>
> Secondly, also in Section 5.1, CBD (with Ott et al., 2018’s implementation) drastically outperforms BD (also with Ott et al., 2018’s implementation). The only difference between CBD and BD is the fact that CBD is trained with the extra synthetic data created by the second-stage back-translation.
>
> This evidence clearly shows that the Ott et al., 2018 implementation is not the source of promotion.
>
> The fact that continuing training with new synthetic data on the mature XLM UMT model produces worse results is because of the technical difficulty in continuous training. To reiterate, the current UMT model is already mature and the learning rate is substantially low, which makes the model struggle to learn new knowledge. This leads the model to perform almost the same as the baseline, with both bt_steps and mt_steps in XLM. Furthermore, if we increase the learning rate, it leads to catastrophic forgetting, causing the performance to deteriorate over time. While there might be a way to fix this problem, but that still does not negate the fact the source of promotion of CBD is from the extra synthetic data created by the second stage.
>
> In order to convince you about the source of promotion, we have tried to run around the clock during the last few hours to run the model entirely with XLM implementation, without any use of Ott et al., 2018’s implementation or any better implementation. We used the data created by the XLM models to train the final model with sorely the same XLM implementation. The model is still running and has not converged yet. Here are the preliminary results based on an intermediate checkpoint.
>
> | Model	 | En-Fr
> |-- |--|
> |Baseline |	33.0 |
> | CBD (entirely XLM implementation, not finished yet) |	34.1 (final value 35.1) |
> |CBD (in the paper)	| 35.3  |
>
> However, it is already enough to surpass the baseline by 1.1 BLEU, which shows that the source of promotion does not come from a better implementation (Ott et al., 2018). Based on our experience, we are confident that the converged model will be (almost) same as the Ott et al., 2018 version. This, along with the evidence in Section 5.1, solidly prove that the main source of promotion does not come from a better implementation but from the extra synthetic data created by the second-stage back-translation.
>
> We have tried our absolute best to satisfy your requirements and clarify your doubts. We hope that our effort will make you have a different view about the paper.
>
> ----------------------
> **Reviewer #4 changed the score from 4 to 5 (still reject) without any further elaboration or any change or update to his/her initial review.**
>
> **Among 4 concerns the reviewer raised, he/she did not respond to our response for point 1,3 and 4. Meanwhile, we completely debunked his/her argument with the undeniable experiments on XLM implementation above.**

---

### Decision · Program_Chairs · 2021-01-07
**Final Decision**

**Decision:**

Reject

**Comment:**

This paper proposed an additional training objective for unsupervised neural machine translation (UNMT). They first train two UNMT models and use these models to generate pseudo parallel corpora.  These parallel corpora are used to optimize the UNMT training objective. The experiments are conducted on several language pairs and they also compared with several alternative works.

All the reviewers admit that the proposed method is straightforward and effective. The authors claim that the new training objective is used to enhance the "data diversification". This point has been questioned by the reviewers. Some reviewers are convinced by the response and some still have different opinions.  From my point of view, the proposed method can also be considered as a kind of combination of  (pseudo) supervised NMT and unsupervised NMT.

The presentation and description of its key contributions seem unclear. However, we encourage the authors to modify their paper and we believe this proposed method can inspire the MT community for further research. At the moment, the paper is seen as not yet ready for publication at this time.